# Green Synthesis of Silver Nanoparticles (Ag-NPs) Using *Debregeasia Salicifolia* for Biological Applications

**DOI:** 10.3390/ma16010129

**Published:** 2022-12-23

**Authors:** Jahanzeb Khan, Irsa Naseem, Saiqa Bibi, Shakeel Ahmad, Faizah Altaf, Muhammad Hafeez, Maha M. Almoneef, Khalil Ahmad

**Affiliations:** 1Department of Chemistry, Mirpur University of Science and Technology (MUST), Mirpur 10250, Pakistan; 2Department of Chemistry, University of Azad Jammu & Kashmir, Muzaffarabad 13100, Pakistan; 3College of Physics and Optoelectronic Engineering, Shenzhen University, Shenzhen 518060, China; 4School of Materials Science and Engineering, Georgia Institute of Technology North Avenue, Atlanta, GA 30332, USA; 5Department of Chemistry, Women University Bagh Azad Kashmir, Muzaffarabad 12500, Pakistan; 6Department of Physics, College of Science, Princess Nourah Bint, Abdulrahman University, Riyadh 11671, Saudi Arabia

**Keywords:** silver nanoparticles (Ag NPs), green synthesis, antioxidant activity, antibacterial activities

## Abstract

The synthesis of nanoparticles (NPs) using the green route is environmentally harmonious and cost-effective compared to conventional chemical and physical methods. In this study, the green synthesis of silver NPs was carried out using an extract of *Debregeasia salicifolia*. The synthesized Ag NPs were characterized by means of different techniques i.e., UV-Vis spectroscopy, FTIR spectroscopy, SEM, and XRD. The XRD pattern exhibited distinctive Bragg’s peaks at (200), (111), (311), and (220). The XRD analysis confirmed the face-centered cubic geometry of the synthesized NPs and revealed that the nature of these NPs is crystalline. The synthesized NPs were verified for their antibacterial activities against Gram-negative *Escherichia coli* (*E. coli*) and Gram-positive *Staphylococcus aureus* (*S. aureus*) bacteria. It showed that antibacterial activity of synthesized silver (NPs) was increased with increasing concentrations of both calcined and non-calcined NPs. The antioxidant activities of Ag NPs were also determined against ABTS at different concentrations for both calcined and non-calcined Ag NPs. Non-calcined Ag NPs have greater antioxidant activity than calcined Ag NPs. This report has a significant medicinal application, and it might open up new horizons in this field.

## 1. Introduction

Nanoscience and nanotechnology deal with the study and use of extremely small objects, and they can be used in any scientific field, including chemistry, biology, physics, material science, and engineering. Nanotechnology is science, engineering, and technology conducted at the nanoscale, which is about 1 to 100 nanometers [1,2,3,4,5]. When the particle size decreases, the surface-to-volume ratio increases as well as the properties (chemical, physical, biological) of the particles, which are different from bulk materials, such as sheets, powder, or loops [6,7,8]. When all of the dimensions of an object or particle are in the nanoscale range, it is referred to as a nanoparticle. Apart from their materials, NPs vary in dimensions and sizes. NPs are classified according to their physiochemical properties/characteristics, sizes, and morphologies. Catalysis includes heterogeneous and homogeneous catalysis and photocatalysis, biomedicine includes antioxidant/antibacterial agents, electrochromic device ECs include smart windows, supercapacitors, photovoltaic devices, magnetic fluids, and gas sensors; there is also magnetic resonance imaging (MRI) and advanced thermal conductivity [9,10]. Carbon-based NPs, organic NPs, inorganic NPs, metal-based NPs, ceramic-based NPs, polymeric NPs, semiconductor NPs, and lipid-based NPs are various types of NPs [11,12].

Cadmium, aluminum, silver, copper, cobalt, zinc, and lead are among the metals in this category. They also include gold, iron, and copper. Their sizes, as well as their characteristics, such as increased surface areas, pore sizes, surface charge densities, rod- and rotund-shaped forms, shade, and sparkling structures, give them various properties. The properties of NPs are affected by environmental elements, such as air, heat, and moisture [13]. Among other properties, Ag NPs have chemical stability, electrical conductivity, antimicrobial properties, catalytic activities, chemical-sensing properties, photonics, electronics, etc. [14,15]. The most essential applications of silver and silver NPs are in the medical industry; they are used in tropical ointments, for example, to treat infections in burns and open wounds. Silver NPs are shown to have anti-inflammatory, antifungal, antiviral, and antiplatelet properties [16]. Apart from being effective, Ag NPs are still popular choices in antibacterial applications due to their non-toxicity to humans in comparison to other metals. Scarcity, on the other hand, makes them expensive and limits their applications. Moreover, chemical methods are toxic, expensive, and hazardous [17,18]. To address this issue, we used a green method to synthesize Ag NPs, which is more cost-effective and cleaner. It is an alternative approach to the chemical method. Instead of the chemical method, researchers prefer the green synthesis/biological method, which is easy to use and eco-friendly. In the biological method, enzymes, microorganisms, plants, etc., can be used [4,19,20]. However, the use of plants instead of microorganisms is more preferred. Although microorganisms are eco-friendly, they are expensive. Therefore, plants are mostly used because they are abundant as well as cost-effective [1,21,22]. In green synthesis, plants act as reducing agents as well as stabilizing agents due to the presence of secondary metabolites, for example, tannins, phenols, organic acid, flavonoids, vitamins, terpenoids, and saccharides [9,23,24,25]. The plant *Debregeasia Salicifolia* was investigated in this study for the synthesis of Ag NPs. *Debregeasia* is a genus in the Urticaceae family, also known as *Tusyar* or *Tushiyari*. A dioecious, evergreen tall shrub or small tree that grows to heights from 0.5 to 5 m, with stems of dark brown fibrous bark and scabrous young shoots [6]. In this study, the green synthesis of silver NPs was successfully accomplished by means of the extract of *Debregeasia salicifolia*. Calcination of Ag NPs occurred at 450 °C for 2 h.

## 2. Experimental Section

### 2.1. Chemicals and Plant Materials 

Analytical grade chemicals were used in this synthesis. Chemicals used in this project were purchased from Sigma Aldrich (Karachi, Pakistan). Silver nitrate (AgNO_3_) was used as the salt precursor. Distilled water was used to prepare solutions and *Debregeasia Salicifolia* leaf extract was used as the reducing and capping agent.

### 2.2. Preparation of Plant Leaves Extract

*Debregeasia salicifolia* leaves were collected from Muzaffarabad, Azad Kashmir. The collected leaves were washed with tap water and then with distilled water and dried in the shade. For the preparation of the *Debregeasia salicifolia* plant extract, 50 g of fresh *Debregeasia Salicifolia* leaves were added into 1000 mL of distilled water and boiled for a maximum of 1 h at 200 °C (Figure 1). The extract obtained was cooled down and filtered through Wattman filter paper 1, and stored at 4 °C for further use [19].

### 2.3. Biogenic Synthesis of Ag NPs

A total of 50 mM of aqueous solution of AgNO_3_ was prepared by dissolving 8.4 g of AgNO_3_ in 1000 mL of distilled water. The prepared plant extract (20 mL) was assorted with 80 mL of 50 mM aqueous salt solution. For 2 h, the reaction mixture was stirred at room temperature and changed in color from colorless to dark brown, which confirmed the synthesis of the Ag NPs; the prepared Ag NPs were aged for 24 h. The sample was then centrifuged for 15 min, washed, and then dried in an oven at 100 °C. The calcination of Ag NPs took place at 450 °C [19].

### 2.4. Characterization

The following techniques are commonly used to characterized nanoparticles: UV–visible spectrophotometry, transmission electron microscopy (TEM), scanning electron microscopy (SEM), Fourier transform infrared spectroscopy (FTIR), and powder X-ray diffraction (XRD) [10].

#### 2.4.1. UV–Visible Spectroscopy

UV–visible spectroscopy is the most frequently used technique for determining the properties of NPs. For analyzing various metal NPs in the size ranges of 2 to 100 nm, light wavelengths between 300 and 800 nm are commonly utilized. Spectrophotometric absorption studies in the wavelength ranges of 300–450 nm is used to characterize silver NPs [26,27].

#### 2.4.2. Scanning Electron Microscopy (SEM)

Electron microscopy is another extensively used approach for the characterization of NPs. Scanning electron microscopy can be used to characterize morphology at the nanoscale to micrometer scale. It describes the shapes and sizes of NPs [28].

#### 2.4.3. Fourier-Transform Infrared Spectroscopy

The surface chemistry can be better understood via FTIR spectroscopy. FTIR can identify organic functional groups (e.g., carbonyls, hydroxyls) linked to the surfaces of NPs as well as other surface chemical residues. It is because, due to vibrations, the subatomic particles in a molecule do not remain in their initial positions and transit to another. If the dipole moment changes on a regular basis, the mode of vibration is infrared (IR), which is activated by molecular vibrations. Each functional group has a wide range of vibrational frequencies and is sensitive to the physiochemical environment; thus, it can provide useful information about the presence of specific functional groups in a sample [8].

#### 2.4.4. X-ray Diffraction Analysis (XRD)

Using the XRD technique, characterization, and phase identification of crystal structures of NPs may be accomplished. The resulting diffraction pattern is created by X-rays penetrating deep into the nanomaterial. The X-ray diffraction analysis was designed for the investigation of crystalline materials. It gives records about crystallinity, crystal alignments, stain phase, crystal imperfections, and structures. X-ray diffraction is basically based on a tester and a positive interference of a monochromatic X-ray. By using a beam tube, these x-rays are generated. Incident rays intermingle with the model and form productive interference when conditions satisfy the Bragg’s law:nλ = 2dsin*θ*
where d is the inter-planar spacing, λ is the wavelength of X-rays, n is an integer, and *θ* is the diffraction angle [14,15].

### 2.5. Antibacterial Activity of Ag NPs

Antibacterial actions of the produced non-calcined and calcined Ag NPs were studied against specific bacteria using the agar well diffusion method. The bacteria used was obtained from biotech, from the lab of the zoology department, UAJ&K (Department of Biotechnology, Muzaffarabad, Azad Kashmir, Pakistan). Gram-negative and Gram-positive bacteria were used during this study. The nutrient agar was used to culture the bacteria. The overnight bacterial culture mixed in a newly prepared, sterilized agar medium was poured into sterilized petri plates and allowed for solidification in a laminar flow at room temperature. By use of a disinfected/sterilized micropipette tip, wells of a 5 mm diameter were made in each plate, and the culture medium plug was removed with a sterilized needle. Different suspensions of the Ag NPs were prepared in deionized water by ultrasonic dispersion and each suspension was shifted into each well and incubated at 37 °C overnight. The zone of inhibition in millimeters (mm) around each well (for the activities of Ag NPs) was checked after 24 h.

### 2.6. Antioxidant Activity

The ABTS free radical scavenging assay was performed to study the antioxidant activities of non-calcined and calcined NPs of Ag. This assay focuses on the decolorization method that takes place when the radical cation ABTS**^˙^**^+^ is reduced to ABTS (2,2′-azinobis-(3-ethylbenzothiazoline-6-sulphonic) diammonium salt). For preparation of the ABTS stock solution, 2.5 mM of potassium persulphate and 7 mM of ABTS were mixed and then kept in the dark for 16 h to generate ABTS**˙**^+-^free radicals. The absorbance (A_o_) of the given solution at 734 nm was recorded by a UV-double beam spectrophotometer. Each calcined and non-calcined sample was dissolved in DW with a concentration of 1 mg /mL to prepare the solution of Ag NPs. For the test, 1 mL of ABTS^˙+^ solution was mixed with the Ag NPs in a range of concentrations, i.e., 1, 3, 5, and 7 mg/mL. The absorbance of sample (A_i_) was nearly observed at 734 nm. Equation (1) is used to calculate the percentage radical scavenging activity, where A_i_ is the absorbance of the test and A_o_ is the absorbance of the control.
%RSA = [(A_o_ + A_i)_/A_o_)] × 100(1)

## 3. Results and Discussion

### 3.1. UV-Visible Spectroscopy

UV-visible spectroscopy was used to examine a colloidal solution of the Ag NPs synthesized by *Debregeasia salicifolia* leaf extract. Figure 2 depicts the UV-visible spectrum. The absorbance maxima for synthesized NPs in the given spectrum are at 470 nm. The presence of a peak in the 300–500 nm range indicated the synthesis of silver NPs [29].

### 3.2. FT-IR Spectroscopy

The chemical compositions of the surfaces of silver NPs were probed by using an FT-IR spectroscopy for the capping of bio-reduced Ag NPs generated by leaf extracts and the reduction of Ag^+^ ions; probable biomolecules that were responsible were identified. Different peaks may be seen in the FTIR spectra of biosynthesized Ag NPs (Figure 3). The peak in the FTIR spectrum at 3591–3269 cm^−1^ reveals the O-H group in alcohols, phenols, and the N-H stretching vibration of protein amides [28]. The -CC- band of alkynes was responsible for the peak at 2125 cm^−1^. The -C = C- band of alkenes caused a peak at 1647 cm^−1^. The C-N stretching vibrations of aliphatic amines might be assigned to the intense band at 1051 cm^-1^. The FT-IR spectrum also exhibited amide I and amide II bands at 1556 and 1371 cm1, which were caused by carbonyl (C = O) and amine (NH) stretching vibrations in the amide links of the proteins, respectively. The C-O stretching of alcohols and carboxylic acids occurred [29]. CH out of plane-bending vibrations of substituted ethylene systems -CH = CH- had peaked at nearly 680 cm^−1^. The positions of the absorption bands in the FTIR spectrum of the plant extract and biosynthesized Ag NPs differed only slightly. The presence of the plant extract compounds in the NP production was established by the shifting of peaks. As a result, plant extract molecules containing OH and CO groups are significant in Ag NP reduction and stability [17].

### 3.3. X-ray Diffraction (XRD) Analysis

The crystalline character of biogenic NPs was inveterate by using the X-ray crystallography method verified on Panalitycal X’pert Pro MRD X-ray diffraction equipment (Malvern Panalytical Ltd., Enigma Business Park, Grovewood Road, Malvern, WR14 1XZ, UK) with Cu K radiation (0.15418 nm) over the scanning range 2 = 30°–80°, with a 0.02 degree step. The crystal structure and phase purity of produced NPs were examined by using this XRD analysis. The XRD pattern in Figure 4 displays very sharp peaks, indicating that the produced NPs were very crystalline.

The XRD pattern of the produced Ag NPs (Figure 4) displays many peaks, with the four main peaks positioned at 38.10°, 44.20°, 64.41°, and 77.39°, respectively, corresponding to the facets of the face-centered cubic (fcc) crystal structure of silver (111), (200), (220), and (311). An intense peak at 32.15° could be attributed to a cubic structure of the material [30]. The crystalline size ‘D’ was 38.15 nm, and calculated by the given equations.
D = *kλ*/ *βcosθ*_B_(2)

The well-known Debye–Scherrer equation is used in Equation (2) to estimate the size of Ag NPs (the average crystallite size). Here, *k* is an empirical constant that depends on the crystallite shape and has a value of 0.89 or 0.9, an X-ray wavelength of 0.15406 nm/1.54, the FWHM (full width at half maximum) is 0.1968, and B is the Bragg angle.

### 3.4. Scanning Electron Microscopy (SEM)

Morphological studies of Ag NPs were presented through a SEM investigation [8]. The morphology of the produced NPs was determined using scanning electron microscopy (SEM) (Thermo Fisher Scientific, 168 Third Avenue, Waltham, MA, USA). Figure 5 shows a 5 µm magnification SEM picture of produced Ag NPs. Figure 5 shows spherical and nearly spherical shapes of Ag NPs, which were agglomerated in some regions. The clustered form of the particles is evident in the SEM pictures, and particles were primarily cuboidal.

### 3.5. Antibacterial Activity of biogenic Ag NPs

Ag NPs have the potential to attach through electrostatic forces of attraction with the surfaces of toxic ions releasing pathogenically and releasing the reactive oxygen species (ROS), which destroys the development as well as growth of these bacteria. Therefore, the agar well diffusion method was used to test the antibacterial activity of Ag NPs extracted from *Debregeasia Salicifolia* against Gram-negative and Gram-positive bacteria. 

Around each well, the zone of inhibition in millimeters (mm) was measured to check the antibacterial activity of calcined and non-calcined Ag NPs. The inhibitory effects of biosynthesized Ag NPs with four different concentrations (20, 40, 60. and 80 mg) were analyzed, as shown in Figure 6a,b and Figure 7a,b. Bacteria i.e., Gram-positive (*S. aures*), and Gram-negative (*Escherichia coli*) were used in the analysis [4,20]. The results show that by increasing the concentrations of suspensions in each well, the antibacterial activities of Ag NPs also increased. The strongest antibacterial activity was observed in the case of non-calcined Ag NPs against (*S. aures)* Gram-positive bacteria. While in the case of calcined Ag NPs (*E. coli*), Gram-negative bacteria were found to be the strongest strain. NPs with large surface areas perform specific functions at the surfaces of microorganisms [3] (shown in Table 1).

Accumulation of these NPs disorganizes the membranes of the microorganisms. The resultant metal ions are responsible for antibacterial activity because of attachments with the surfaces of the microorganisms. Antibacterial activity is typically dependent on the nanoparticle size, the large surface area, reactive radicals (Ag^+^), and reactive oxygen species (ROS). 

The NPs of Ag are acknowledged antibiotics that have the capacity to yield ROS, such as hydroxyl radicals, peroxide, superoxide, and singlet/alpha oxygen, following the Fenton reaction [20]. These ROS—when attacking bacterial/pathogen surfaces—cause damage to the DNA, oxidation of the protein, peroxidation of the lipid, and death to the bacterial cell. (Shown in Figure 8).

### 3.6. Antioxidant Activity of Biogenic Ag NPs

Biogenically synthesized Ag NPs act as good antioxidants and block the latent oxidation reduction of numerous species. The radical scavenging potential of these NPs makes them scavengers against reactive oxygen species (ROS). Ag NPs are excellent oxidative stress inhibitors, which may be radical or non-radical in nature. Various methods are taken/selected for antioxidant activity, but the ABTS free radical assay is considered one of the best and most reliable methods. The ABTS free radical assay is also known as the Trolox equivalent antioxidant capacity (TEAC) method. 

In this method, the ABTS radical scavenging assay was used to reduce the free radical ABTS^˚+^ cation into ABTS (2,2′-azinobis-(3-ethylbenzothiazoline-6-sulphonic) diammonium salt [27]. Figure 9 shows the antioxidant activities (ABTS free radical scavenging) of biogenic non-calcined Ag NPs. The ABTS free radical scavenging assay was performed at different amounts (1, 3, 5, and 7 mg/mL) of Ag NPs. The strong ABTS scavenging power of non-calcined (84%) was recorded at 7 mg/mL for biogenic Ag NPs. The ABTS test demonstrated the latent scavenging activity of Ag NPs for reactive oxygen species (ROS). The following studies show that the aqueous solution of *Debregeasia Salicifolia* operated as a good reducing oxidizer, as well as a capping agent. There were plants in studies that contained terpenoids, polar flavonoids, non-polar phenols, flavonoids, saponins, tannins, and ascorbic acid with different capacities to scavenge the role of free radicals. The plant is a good antioxidant type that also improves the oxidation reduction potential of Ag NPs. The resultant NPs showed distinctive antioxidant potential and reduced the oxidative stress. Figure 9 shows the straight-line equation (y = 10.85x + 14.35), which explains that, upon increasing the concentration of biogenic Ag NPs, the percentage scavenging activity of Ag NPs also increases. 

The IC_50_ value of non-calcined Ag NPs was 3.28 mg/mL. Figure 9 shows the antioxidant activities (ABTS free radical scavenging) of biosynthesized calcined Ag NPs. The ABTS free radical scavenging assay was performed at different concentrations (1, 3, 5, and 7 mg/mL) of Ag NPs (tabulated in Table 2).

The strong ABTS scavenging power of the calcined Ag NPs (81%) was recorded at 7 mg/mL (for biogenic Ag NPs). Figure 9 shows the straight-line equation (y = 10.6x + 13.1), which explains that, upon increasing the concentration of biogenic Ag NPs, the percentage of the scavenging activity of Ag NPs also increases. The IC_50_ value of uncalcined Ag NPs was 3.48 mg/mL. From the given results, it is concluded that non-calcined Ag NPs had more antioxidant activities than calcined Ag NPs because non-calcined Ag NPs had various compounds that might have participated in the reduction and stabilization of biogenically synthesized Ag NPs via *Debregeasia Salicifolia*. The percentage of radical scavenging activity was determined by the formula given below. In Equation (3), A_i_ is the absorbance of the test and A_o_ is the absorbance of the control.
%RSA = [(A_o_ + A_i_)/A_o_)] × 100(3)

## 4. Conclusions

*Debregeasia salicifolia* leaf extract was used to synthesize silver NPs using a green method. These NPs were characterized, and their antibacterial and antioxidant properties were demonstrated. A two-hour calcination process was also carried out after the formulation of these NPs. By using several characterization techniques, the synthesized NPs were confirmed. The progress of the reaction was monitored by UV-visible measurements. Different biological substances that stabilized and capped the silver NPs were identified using FTIR measurements. The crystalline character of synthesized Ag NPs was revealed by XRD measurements. The morphology of the assembled NPs, as assessed by SEM, was the face-centered cubic. The produced Ag NPs demonstrated strong antibacterial action against Gram-positive (*S. aureus*) and Gram-negative (*E. coli*) bacteria, and this antibacterial activity was enhanced by increasing the concentration of the NPs. Ag NPs that were not calcined had stronger antioxidant activity (3.28 mg/mL) than those that had (3.48 mg/mL) for the non-calcined. This study provided proof that *Debregeasia salicifolia* can be used to make silver nanoparticles. Finally, it can be said that *Debregeasia salicifolia* is a superior candidate for the fabrication of silver NPs using an easy and environmentally friendly method.

## Figures and Tables

**Figure 1 materials-16-00129-f001:**
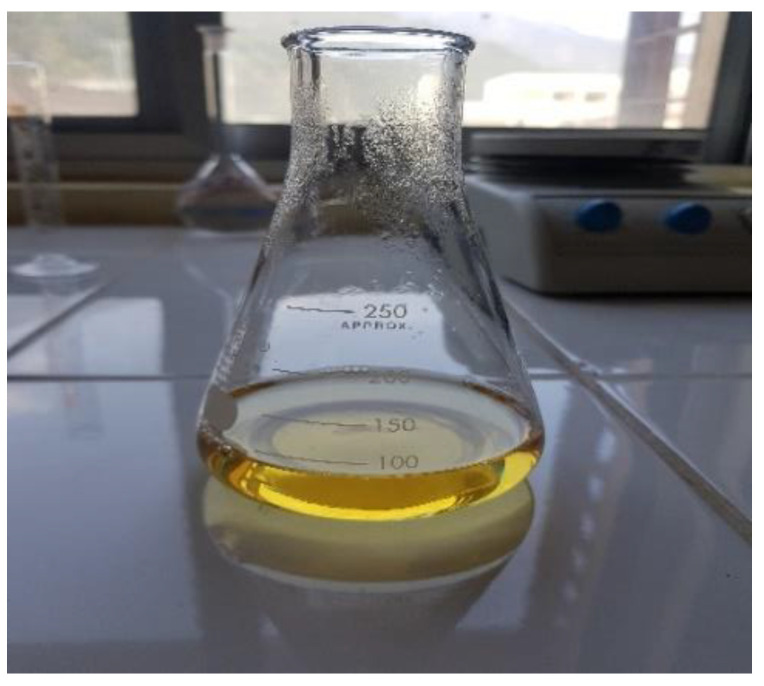
Debregeasia salicifolia leaf extract.

**Figure 2 materials-16-00129-f002:**
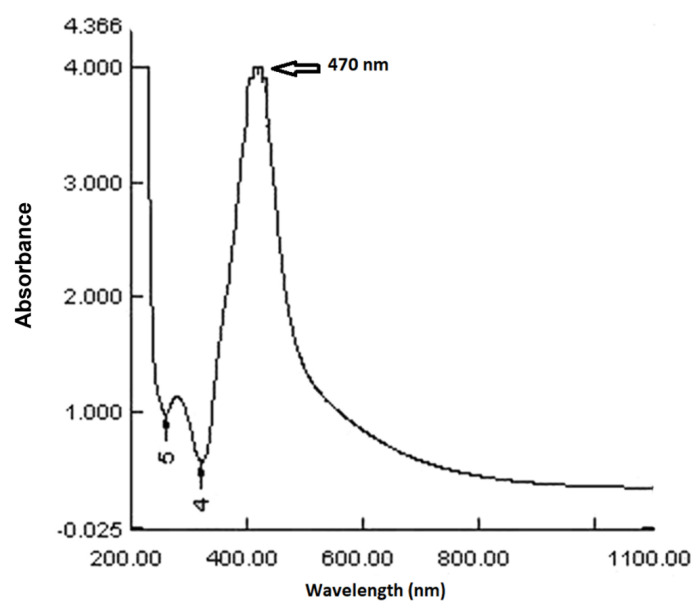
UV-visible spectrum of greenly synthesized Ag NPs.

**Figure 3 materials-16-00129-f003:**
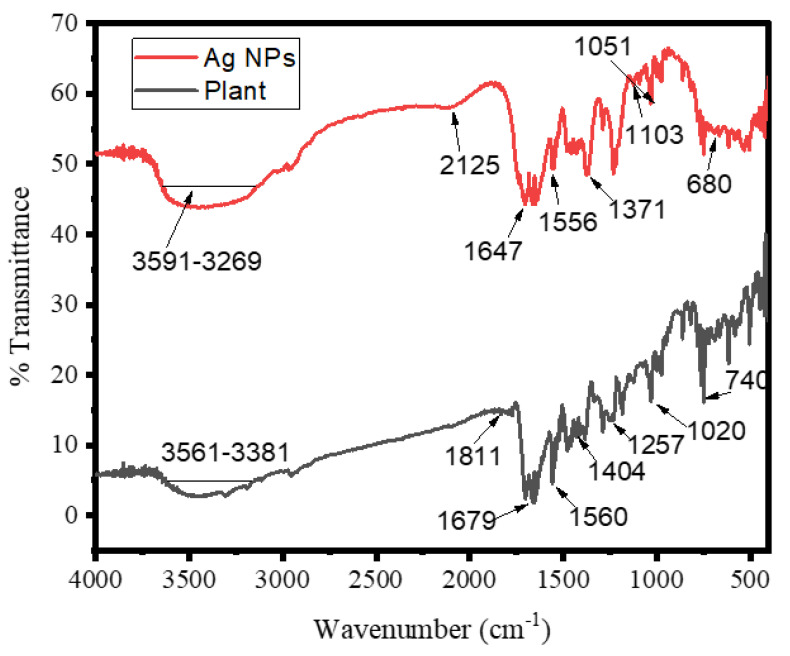
FTIR spectrum of aqueous extract of *Debregeasia Salicifolia* leaves and Ag NPs at room temperature.

**Figure 4 materials-16-00129-f004:**
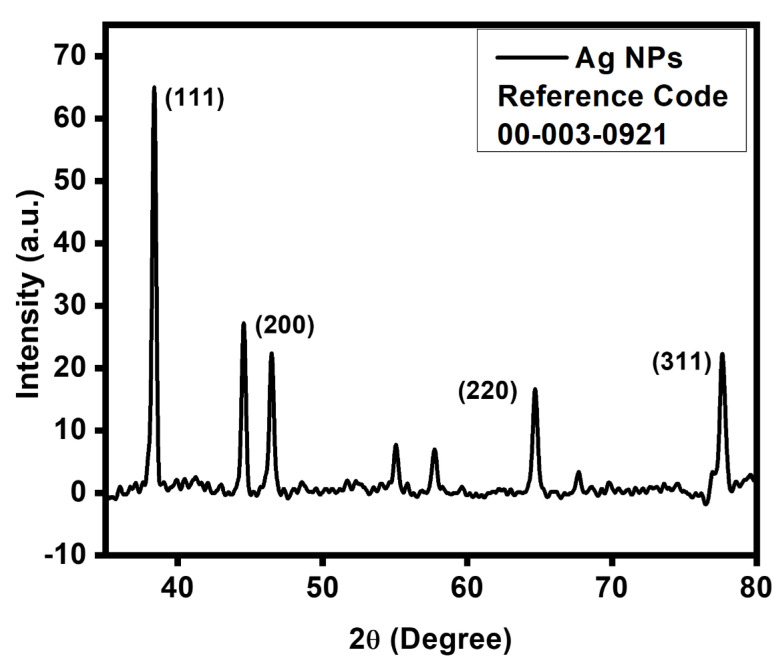
XRD pattern of biogenic Ag NPs.

**Figure 5 materials-16-00129-f005:**
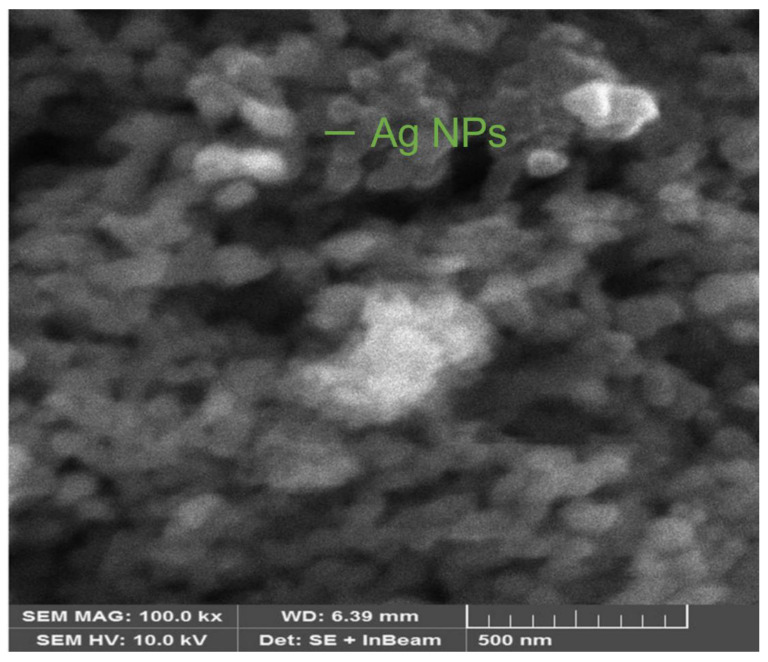
Scanning electron microscopy of biogenic Ag NPs.

**Figure 6 materials-16-00129-f006:**
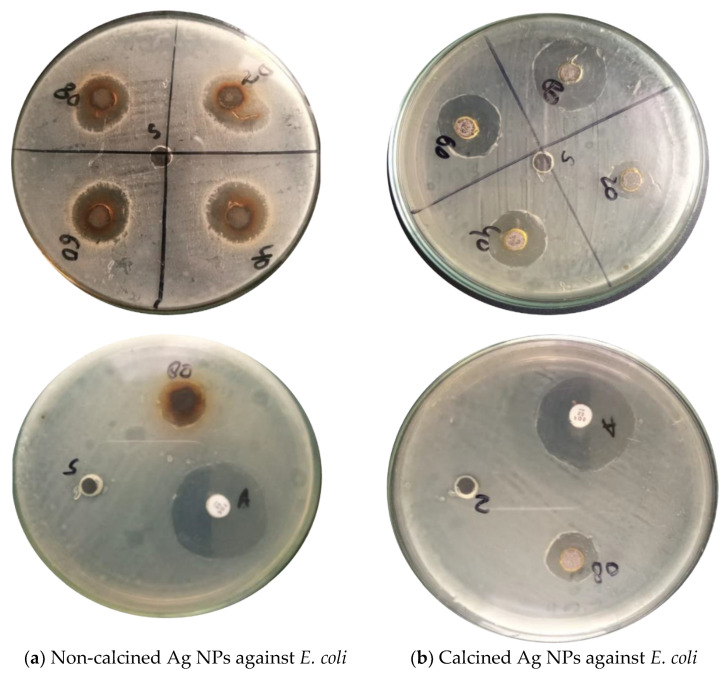
Antibacterial activity of biosynthesized (**a**) Non-calcined Ag NPs against *E. coli*, (**b**) Calcined Ag NPs against *E. coli*.

**Figure 7 materials-16-00129-f007:**
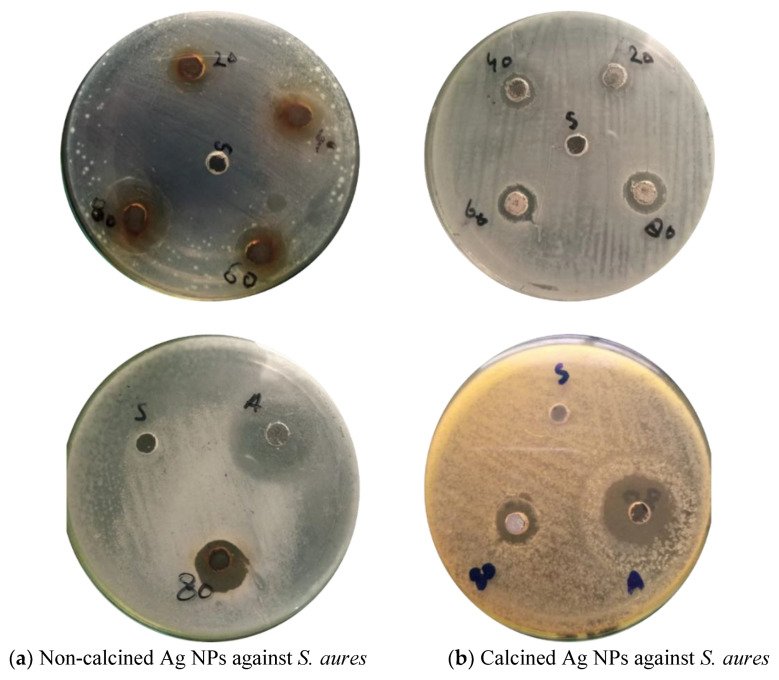
Antibacterial activity of biosynthesized (**a**) Non-calcined Ag NPs against *S. aures,* (**b**) Calcined Ag NPs against *S. aures*.

**Figure 8 materials-16-00129-f008:**
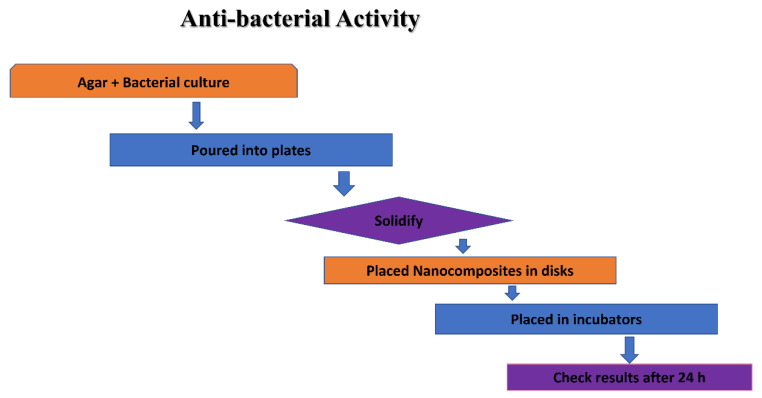
Schematic representation of the antibacterial activity of green-synthesized Ag NPs.

**Figure 9 materials-16-00129-f009:**
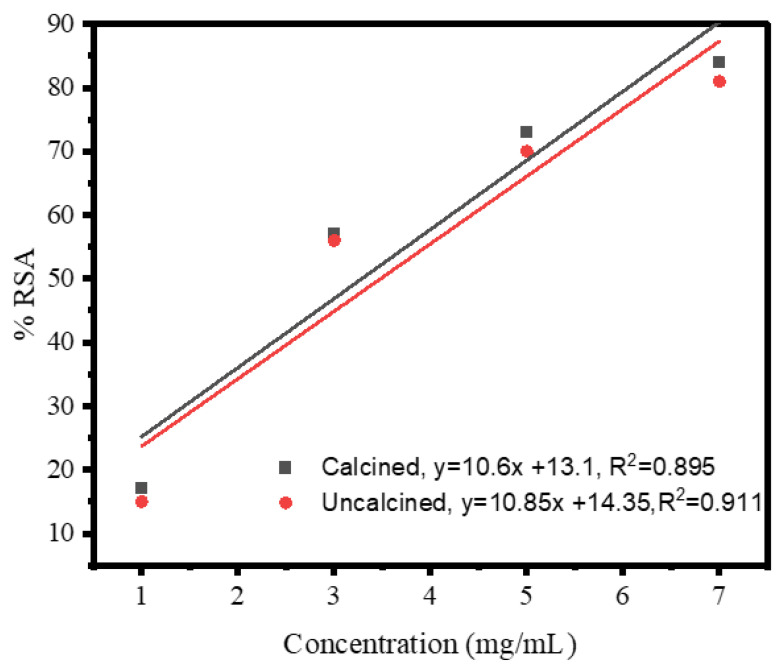
Graphical representation of antioxidant activity of green-synthesized calcine and non-calcined Ag NPs.

**Table 1 materials-16-00129-t001:** Antimicrobial activities of Ag NPs against different microbes, where Al = 20 mg/mL, A2 = 40 mg/mL, A3 = 60 mg/mL, and A4 = 80 mg/mL.

**Zone of Inhibition (ZOI) Measure in Millimeters (mm)**						
	**Standard**	**Solvent Activity**	**A1**	**A2**	**A3**	**A4**
Bacteria for Non-calcined Ag NPs						
*Escherichia coli*	29	00	10	15	16	20
*S. aures*	29	00	14	15	18	21
Bacteria for Calcined Ag NPs						
*Escherichia coli*	22	00	15	17	20	22
*S. aures*	22	00	8	10	12	13

**Table 2 materials-16-00129-t002:** Percentage scavenging activity (%RSA) of biogenic non-calcined and calcined Ag NPs and IC_50_ from the ABTS assay.

Sample	Concentration mg/mL	Ao	A_i_	%RSA	IC_50_
Non-calcined Ag NP_S_	1	0.1	0.083	17	3.28
	3	0.1	0.043	57	
	5	0.1	0.027	73	
	7	0.1	0.016	84	
Calcined Ag NP_S_	1	0.1	0.085	15	3.48
	3	0.1	0.044	56	
	5	0.1	0.03	70	
	7	0.1	0.019	81	

## Data Availability

Not applicable.

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
