# Peer review of "Green Synthesis of Silver Nanoparticles (Ag-NPs) Using Debregeasia Salicifolia for Biological Applications"

_materials, 2022, doi:10.3390/ma16010129_

Round 1
Reviewer 1 Report (Previous Reviewer 1)
I recommend accepting after minor revision, I have highlighted in the pdf attached. and this improved version is better than before

Author Response
Reviewer 1
I recommend accepting after minor revision, I have highlighted in the pdf attached. and this improved version is better than before
Thanks for your valuable feedback. We've worked hard to improve our manuscript's overall quality, as well as its language, typos, section and numbering mistakes, and figure quality. We have accommodated your all suggestions in our final file of the manuscript and highlighted them respectively. (All suggestions highlighted in the pdf file have been accommodated). Now, we are hopeful to publish this work in this journal in its current form.

Reviewer 2 Report (Previous Reviewer 4)
Dear Authors,
The revised version of the manuscript entitled: "Green Synthesis of Silver Nanoparticles (Ag-NPs) Using Debregeasia Salicifolia for Biological Applications” is much improved than the previously submitted report and can be considered for publication in Materials.
Author Response
Reviewer 2
Dear Authors,
The revised version of the manuscript entitled: "Green Synthesis of Silver Nanoparticles (Ag-NPs) Using Debregeasia Salicifolia for Biological Applications” is much improved than the previously submitted report and can be considered for publication in Materials.
Thanks for your valuable feedback. We've worked hard to improve our manuscript's overall quality, as well as its language, typos, section and numbering mistakes, and figure quality. Now, we are hopeful to publish this work in this journal in its current form.

Reviewer 3 Report (New Reviewer)
Comments to the Authors
In this manuscript authors prepared silver NPs using an extract of Debregeasia salicifolia and The synthesized NPs were verified 30 for their antibacterial activities against gram negative Escherichia coli (E. coli) and gram-31 positive Staphylococcus aureus(S. aureus) bacteria. This research has value for the researchers in the related areas. However, the paper needs improvement before acceptance for publication. My detailed comments are as follow:
1. In the introduction section Authors describe catalytic properties of AgNPs through this sentences “Catalysis (heterogeneous and homogeneous catalysis, photocatalysis)……….…..” After this sentence authors should provide relevant references as mention below:
a. doi.org/10.1007/s40089-021-00362-w
b. doi.org/10.1002/slct.201900470
2. The quality of the Figure 3.1 should be improved and peak of AgNPs should be marked in the spectra.
3. Authors should mention the peaks with reference code in the text along with appropriate references.
4. Authors should mark the AgNPs in the SEM images.
5. There are few typos errors authors should correct it.
Authors should provide a schematic mechanism diagram of Anti-microbial activity.
Author Response
Reviewer 3
In this manuscript authors prepared silver NPs using an extract of Debregeasia salicifolia and The synthesized NPs were verified 30 for their antibacterial activities against gram negative Escherichia coli (E. coli) and gram-31 positive Staphylococcus aureus (S. aureus) bacteria. This research has value for the researchers in the related areas. However, the paper needs improvement before acceptance for publication. My detailed comments are as follow:
- In the introduction section Authors describe catalytic properties of AgNPs through this sentence “Catalysis (heterogeneous and homogeneous catalysis, photocatalysis) ………. …..” After this sentence authors should provide relevant references as mentioned below:
- doi.org/10.1007/s40089-021-00362-w
- doi.org/10.1002/slct.201900470
We have added these references in the main file of our manuscript at page # 2 and line # 53.
- The quality of the Figure 3.1 should be improved and peak of AgNPs should be marked in the spectra.
We have inserted an improved resolution / quality figure in the main file of our manuscript at page # 7 and line # 170-74.
- Authors should mention the peaks with reference code in the text along with appropriate references.
We have mentioned the peak position with reference in the main file of our manuscript at page # 7 and line # 169-70.
- Authors should mark the AgNPs in the SEM images.
We have marked the Ag NPs in the SEM image in the main file of our manuscript at page # 10 and line # 217-19.
- There are few typos’ errors authors should correct it.
Authors should provide a schematic mechanism diagram of Anti-microbial activity.
We've worked hard to improve our manuscript's overall quality, as well as its language, typos, section and numbering mistakes, and figure quality. We have added a schematic mechanism diagram of the anti-microbial activity diagram in the main file of our manuscript on page # 12.

This manuscript is a resubmission of an earlier submission. The following is a list of the peer review reports and author responses from that submission.
Round 1
Reviewer 1 Report
Review Report
I would like to thank all authors of the manuscript for their good and novelty manuscript titled as (Green Synthesis of Silver Nanoparticles (Ag-NPs) Using DEBREGEASIA SALICIFOLIA for Biological Applications which submitted to journal (Materials (ISSN 1996-1944).
1-The manuscript is original and novel as it aims to Synthesis of nanoparticles (NPs) using green route is environmentally harmonious and cost effective and their application against gram positive (S.aures) and gram negative (Escherichia coli) Antioxidant activity of AgNPs was also determined against ABTS at different conc.
2-The Presentation of the manuscript is good which attract the Interest to the readers.
3-Minor revision is needed to English language and style.
4-The introduction provide sufficient background and include all relevant references and styled according to the style of the journal.
5- All the cited references relevant to the research.
6- All the cited references relevant to the research.
7- The methods adequately described.
8- The results clearly presented.
9- The conclusions supported by the results.
So, I recommend accepting after minor revision (corrections to minor methodological errors and text editing.
Corrections are:
In keywords
Corrections are highlighted in the page 2.
In Introduction
Corrections are highlighted in the page 2 ,3
In Materials and Methods
Corrections are highlighted in the page 4 ,5,6
In Results
Corrections are highlighted in the page 6,7,8,9,10,11,12,15
Reviewer 2 Report
In this manuscript entitled “Green Synthesis of Silver Nanoparticles (Ag-NPs) Using DEBREGEASIA SALICIFOLIA for Biological Applications” (Manuscript Number: materials-1928646) I think it’s better to discuss about below questions. Therefore, I suggest a major revision for the manuscript. Comments:
1. Experimental: should be extended. Also, the total materials and companies, characterization of all instruments, methodology and synthesis steps should be presented with more details.
2. All equations should be referred in the text by a number.
3. There are many papers about green synthesis of silver nanoparticles. The authors should be explaining about the importance and novelty of the work with more details.
4. A comparison table should be presented to show novelty of the work.
5. The English language should be improved (Example: by or using instead by using and …).
6. The grammatical errors should be corrected.
Reviewer 3 Report
Abstract:
Line 3- Replace the word "synthesized" by "is carried out using an extract of"
Line 4- 450 C0 is wrong, it should be 4500C.
Throughout the manuscript, hours should be written as "h", minutes as "min" and seconds as "s".
Line 5- delete "instrumental"
Line 9- S. aureus write complete name at first appearance after that you can use this as S. aureus throughout the manuscript. Escherichia coli, first time full name, after that E.coli throughout the namuscript.
Line 10- add "of" before both.
Line 11- it is better to write non-calcined than uncalcined.
INTRODUCTION
Line 2: write scientific fields instead of science field.
Line 3: Rewrite this sentence "The operation of stuff with at least one aspect size ranging from 1 to 100 nm is referred to as nanotechnology.[1-5]"
Line 4: Replace "particles" with particle.
The paragraph "Metals in this category include cadmium, aluminum, silver copper, cobalt, zinc, and lead gold, ........
is too sudden. There should be a flow in your writing.
In this study, green synthesis of silver NPs is synthesized by means of extract of Debregeasia Salicifolia. Calcination of AgNPs was done at 450 CËš for two hours.
replace synthesized in the above sentence by "is carried out using the extract of" correct 450 0C.
2.2 Preparation of plant leaves extract
Debregeasia Salicifolia leaves collected from Muzaffarabad,
In the above sentence add "were collected"
The extract obtained was cooled down & filtered by Wattman filter paper 1,
replace "by" with "through"
avoid short form of the words which are not standard for example (soln, conc, etc)
This Manuscript needs language corrections throughout. The authors should check each line carefully and correct it to make it understandable.
The SEM image is not so clear. Please provide the clearer/better image.
Reviewer 4 Report
The term “DEBREGEASIA SALICIFOLIA” in title should be changed Debregeasia salicifolia.
Remove the (-) symbol from the Ag-NPs in throughout manuscript.
The author should provide some quantitative information in the abstract section.
The author should correct the spelling in S. aureus instead of S. aures in throughout manuscript.
In the experimental section 2.4.5 Anti-bacterial activity of Ag NPs and 2.4.6 Anti-oxidant activity should be changed the section number like this, 2.5 Anti-bacterial activity of Ag NPs and 2.6 Anti-oxidant activity.
Overall this manuscript, all the figure qualities are poor. UV-Visible spectra is drawn by yourself. I couldn’t read this manuscript carefully, due to the text and figure numbers not provide properly. This lack of this performance, I couldn’t recommendation this paper in this Journal.
Reviewer 5 Report
In the manuscript (nanomaterials-1943934), “Green Synthesis of Silver Nanoparticles (Ag-NPs) Using DEBREGEASIA SALICIFOLIA for Biological Applications” by J. Khan, I. Naseem, S. Bibi, S. Ahmad, F. Altaf, M. Hafeez, M.M. Almoneef, and K. Ahmad authors synthesized silver nanoparticles (Ag NPs) using seed extract of Foeniculum vulgare, followed by an examination of their antioxidant and antibacterial activities against gram-negative (Escherichia coli) and gram-positive (S. aureus) bacteria. Before antioxidant and antibacterial tests, prepared samples were characterized by UV-Vis and FTIR Spectroscopy, SEM, and XRD. However, the manuscript does not carry much novelty, is written confusingly, and is even technically unsatisfactory. Because of that, I do not recommend the manuscript for publication in Materials.
First, a significant effort of the scientific community exists to prepare Ag NPs with plant extracts. The introduction does not connect the presented results/intents with previous work in this field but describes common knowledge concerning nanomaterials.
Second, instead of providing information about the equipment in the Experimental section concerning characterization, the authors describe what information those experimental techniques can provide. For example, the first sentence in Results and Discussion (Section 3.3) should be in the Experimental section.
Concerning characterization, what kind of samples are characterized, calcinated or un-calcinated, and is there any difference?
The absorption spectrum has an unusual shape. The authors stated (page 7): “The absorbance maxima for Synthesized NPs in the given spectrum are at 302 nm.”, but the peak is at 470 nm.
SEM is of poor quality and non-informative. The authors stated (page 9): “Figure 4.5 shows a 5 µm magnification.” However, the magnification is 5000, while the bar is 5 µm.
Figure numeration between the text and the figures is inconsistent. For example, figure 4.1 (text) and figure 3.1 (figure caption), figure 4.2 (text) and figure 3.2 (figure caption), etc.
There are spelling mistakes. For example, frequently, instead of silver is “sliver”. Significant improvement in English is necessary.
